# The Novel Strategy for Increasing the Efficiency and Yield of the Bipolar Membrane Electrodialysis by the Double Conjugate Salts Stress

**DOI:** 10.3390/polym12020343

**Published:** 2020-02-05

**Authors:** Dong Wang, Wenqiao Meng, Yunna Lei, Chunxu Li, Jiaji Cheng, Wenjuan Qu, Guohui Wang, Meng Zhang, Shaoxiang Li

**Affiliations:** 1College of Environment and Safety Engineering, Qingdao University of Science and Technology, Qingdao 266042, China; wangdong@qust.edu.cn (D.W.); wenqiaoqust@163.com (W.M.); leiyunna207@163.com (Y.L.); 15588685658@163.com (J.C.); qwj_710@163.com (W.Q.); 17864298538@163.com (G.W.); A903962566@163.com (M.Z.); 2Shandong Engineering Research Center for Marine Environment Corrosion and Safety Protection, Qingdao University of Science and Technology, Qingdao 266042, China; 3Shandong Engineering Technology Research Center for Advanced Coating, Qingdao University of Science and Technology, Qingdao 266042, China; 4ASTUTE 2020 in Future Manufacturing Research Institute, College of Engineering, Swansea University, Swansea SA1 8EN, UK

**Keywords:** BMED, current efficiency, conjugate salt, influence factor

## Abstract

To improve sulfuric acid recovery from sodium sulfate wastewater, a lab-scale bipolar membrane electrodialysis (BMED) process was used for the treatment of simulated sodium sulfate wastewater. In order to increase the concentration of sulfuric acid (H_2_SO_4_) generated during the process, a certain concentration of ammonium sulfate solution was added into the feed compartment. To study the influencing factors of sulfuric acid yield, we prepared different concentrations of ammonium sulfate solution, different feed solution volumes, and different membrane configurations in this experiment. As it can be seen from the results, when adding 8% (NH_4_)_2_SO_4_ into 15% Na_2_SO_4_ under the experimental conditions where the current density was 50 mA/cm^2^, the concentration of H_2_SO_4_ increased from 0.89 to 1.215 mol/L, and the current efficiency and energy consumption could be up to 60.12% and 2.59 kWh/kg, respectively. Furthermore, with the increase of the volume of the feed compartment, the concentration of H_2_SO_4_ also increased. At the same time, the configuration also affects the final concentration of the sulfuric acid; in the BP-A-C-BP (“BP” means bipolar membrane, “A” means anion exchange membrane, and “C” means cation exchange membrane; “BP-A-C-BP” means that two bipolar membranes, an anion exchange membrane, and a cation exchange membrane are alternately arranged to form a repeating unit of the membrane stack) configuration, a higher H_2_SO_4_ concentration was generated and less energy was consumed. The results show that the addition of the double conjugate salt is an effective method to increase the concentration of acid produced in the BMED process.

## 1. Introduction

With the development of economy and industrialization, a great deal of wastewater is generated during industrial development. Sodium sulfate, as a commonly used industrial raw material, is also one of the main components of most industrial wastewater. The discharge of a large amount of industrial wastewater that contains sodium sulfate not only pollutes water sources, but also causes salinization of the soil and damages the ecological environment. Environmental protection has been the focus of attention, and the development of effective technologies for the treatment of sodium sulfate wastewater has become a top priority.

Bipolar membrane electrodialysis (BMED) can dissociate water into hydrogen ions and hydroxide ions under the driven force of potential difference. Acid and base solutions will be produced at the corresponding compartment of a BMED stack. For this case, BMED has been widely used for organic and inorganic acid/base production, environmental protection, and resources isolation, especially for the industrial development of zero emissions [1,2].

In recent years, people have also discovered the superior performance of bipolar membrane electrodialysis compared to traditional technologies, including no gas or by-product generation, a lower voltage drop, maximal energy utilization, space saving, and easier installation and operation.

A series of studies have been carried out on the treatment of wastewater by BMED, especially in the treatment of sodium sulfate wastewater. Paleologou et al. [3] reported that by a two-compartment BMED system at a rate of 46 mL/min of sodium sulfate, wastewater with a current efficiency of 78%, 1.08 mol/L NaOH could be produced.

Kroupa et al. [4] have performed a preliminary economic evaluation of generating NaOH and H_2_SO_4_ from a uranium ore mining wastewater with a bipolar membrane cell. Recently, Zhou et al. [5] recovered H_2_SO_4_ from sodium sulfate through a BMED process. In this process, the current efficiency and energy consumption of produced H_2_SO_4_ could reach 82.7% and 1.97 kWh/kg, respectively. Tufa et al. [6] presented a novel approach combining reverse electrodialysis (RED) and alkaline polymer electrolyte water electrolysis (APWEL) for renewable hydrogen production. Cejna et al. [7] utilized membrane distillation (MD) and membrane crystallization recovered Na_2_SO_4_ crystalline product from wastewater. Arthur et al. [8] utilized electrodialysis with the bipolar membrane (EDBM) to examine and neutralize the desalinated whey after ED. Miao et al. [9] employed bipolar membrane electrodialysis (BMED) for the preparation of TMAdaOH from its halide with a BP-C1-A1-C2-BP four-compartment configuration. Lu et al. [10] proposed that bipolar membrane electrodialysis (BMED) is an eco-friendly method for the preparation of choline hydroxide from choline chloride.

These studies indicated that BMED technology had a good application prospect in the treatment of industrial wastewater and plays an important role in environmental protection.

Although most of the components in the wastewater can be reclaimed by the BMED process, the biggest problem is the low yield of sulfuric acid recovery, which limits the application of BMED in industrial wastewater treatment [11]. To increase the acid concentration, Zhang et al. [12] proposed filling a strong acid resin into the acid chamber to increase the acid concentration and explored the method of the energy-saving producing weak acid by BMED with the three-compartment configuration, thus enhancing the concentration of acid production (from 0.14 to 0.16 mol/L) with relatively high current density and relatively low energy consumption at the current density of 70 mA/cm^2^.

Jian et al. [13] improved water dissociation by loading MoS_2_ nanosheets on bipolar membranes interfaces as advanced catalysis at 90 °C, which exhibited an outstanding performance with a high current efficiency of 45% and a low energy consumption of 3.6 kWh/kg, which increased the acid concentration (from 0.04 to 0.23 mol/L) at a current density of 20 mA/cm^2^.

It can be concluded from the results that the acid concentration increased obviously after lots of effort. However, the bipolar membrane electrodialysis method has not completely solved problems such as the impurities, instability, and feasibility in the practical application of industrial wastewater. Therefore, more research studies should be conducted for increasing the acid concentration by the BMED process.

In this study, simulated sodium sulfate wastewater was treated by BMED for the generation of H_2_SO_4_ and NaOH. The conjugate salt, ammonium sulfate was added in the salt compartment to increase the yield of acid during the process.

In the base compartment, the mixed base solution could be separated by the different volatility of sodium hydroxide and ammonium sulfate. The influence of current density, the concentration of ammonium hydroxide, the volume of wastewater, and the membrane stack structure on the BMED process were investigated. The current efficiency, energy consumption, and acid concentration under different conditions were examined to evaluate the increase of the acid concentration. The optimized conditions of the system were obtained.

## 2. Materials and Methods

### 2.1. Materials

In the experiment, the sodium sulfate, ammonium sulfate, and sodium hydroxide purchased from the Sinopharm Chemical Reagent Co., Ltd. (Beijing, China) were used in the chemical analyses. Deionized water (Millipore Milli-Q 18 MΩ, Bedford, MA, USA) was used throughout. The Selemion homogeneous cation exchange membrane CMV and Selemion anion exchange membrane AMV used during the experiment were purchased from Asahi Glass Co., Suzhou, China and the bipolar membrane (BPM) was provided by Tokuyama Astom Co., Tokuyama, Japan. The main properties of the ion exchange membrane are presented in Table 1.

### 2.2. Experimental Set-Up

The lab-scale experimental set-up was mainly composed of a three-compartment electrodialysis stack, a DC power supply, four 2 L tanks, four centrifugal pumps, and four flow meters. The electrodialysis stack, which was purchased from Shandong Tianwei membrane technology Co., Ltd., Weifang, China, was made up of two electrodes (titanium plate coated with ruthenium), spacers, and ion-exchange membranes. The stack embraced five repeating units, and the style of bipolar membrane configuration in this study was BP-A-C-BP; namely, there were three compartments in each unit, including acid, salt, and base compartments. The effective area of each membrane was 100 cm^2^. The structure of BMED configuration was shown in Figure 1a. The DC power supply (WYL1720, Hangzhou Siling Electrical instrument Ltd., Hangzhou, China) provided direct current field for the stack. The anions (SO_4_^2−^) and cations (Na^+^, NH_4_^+^) in the salt compartment were allowed to migrate through anion exchange membranes and cation exchange membranes respectively into acid and base compartments under the direct current field, and then they were combined with H^+^ and OH^−^ generated from water splitting at the interface of the BPM to form the corresponding acid and base. The solutions were circulated in the three-compartment system driven by the centrifugal pump (CXB-30 Wenzhou Erle Pump Co., Ltd., Wenzhou, China). The structure schematic diagram of the BMED process is shown in Figure 2.

In the experiment, the salt compartment and acid/base compartment were filled with 5 wt % sodium sulfate solution and the deionized water, respectively. The electrode compartment was rinsed with 1 L of 5 wt % Na_2_SO_4_ solution as the supporting electrolyte. The solutions in compartments were driven by the centrifugal pump at a constant flow rate of 12 L/h. The whole experimental process was carried out under constant current conditions, and the temperature of the experimental system was maintained at 25 °C.

### 2.3. Analysis Methods and Calculations

The samples with a volume of 10 mL were taken every 20 min from the acid compartment during the BMED process. The concentration of sulfuric acid was determined by titration with 1 mol/L sodium hydroxide solution (standard) using phenolphthalein as an indicator.

The current efficiency η (%) was calculated as shown in Equation (1):(1)$$\eta =\frac{\left(C_{t}-C_{0}\right)ZFV}{NIt}\times 100\%$$
where *C*_0_ and *C*_t_ (mol/L) are the concentration of acid in the acid compartment at time 0 and *t* (s). *Z* is the ions’ valence (*Z* = 2), *F* is the Faraday constant (96,485 C/mol), *V* is the volume (L) of solution, *N* is the number of repeating units (*N* = 5), and *I* is the current (A).

The energy consumption *E* (kWh/kg) was calculated as shown in Equation (2):(2)$$E=\frac{\int_{t}^{0}U_{t}Idt}{C_{t}VM}$$
where *U*_t_ is the voltage (V) across the membrane stack at time *t* (s), *I* is the current (A), *C*_t_ (mol/L) is the concentration of acid in the acid compartment at time *t* (s), *V* is the volume (L) of the solution, and *M* is acid’s molar mass (*M* = 98.078 g/mol).

## 3. Results and Discussion

### 3.1. Effects of Current Density

In order to explore the influence of current density on the current efficiency, energy consumption, and the concentration of acid before and after the addition of ammonium sulfate, the experiments were performed with 15 wt % Na_2_SO_4_ and 5 wt % (NH_4_)_2_SO_4_ at different current densities (30–70 mA/cm^2^).

The voltage across the membrane stack is an important parameter, which is closely related to the current efficiency and energy consumption. Therefore, the effect of current density on the voltage before and after the addition of 5 wt % (NH_4_)_2_SO_4_ was demonstrated in Figure 3. During the whole experiment at constant current, it can be found that whether (NH_4_)_2_SO_4_ was added or not, the applied voltage declined sharply at the start of the experiment procedure. This was mainly related to the increase of the conductivity of the acid/base compartment caused by the splitting of water molecules and the migration of the hydrate ions, reducing the electrical resistance of the stack. Subsequently, the BMED stack reached a stable state in the intermediate phase for a while, presenting a slight change in the stack voltage, which suggested that the stack resistance remained almost constant in this stage [14]. In the period, the increase of the resistance in the salt compartment, resulting from the continuous decrease of the mixture of Na_2_SO_4_ and (NH_4_)_2_SO_4_ solutions, was counteracted by the reduction of the resistance of the acid and base compartments, since the increase of the H_2_SO_4_ and NaOH concentrations [15]. Due to the transformation of Na_2_SO_4_ and (NH_4_)_2_SO_4_ being completely achieved, the resistance of the feed solution increased, resulting in a sharp increase of the stack voltage at the final phase.

Regardless of whether (NH_4_)_2_SO_4_ was added or not, the voltage will increase as the current density increases. These can be observed in Figure 3. This can be attributed to the variation of voltage and current conformed to Ohm’s law; that is, the larger the current, the higher the voltage, and there will be a relatively constant stack resistance [16,17]. When a higher current was applied, the water dissociation in the interface layer of the bipolar membrane and the transfer of various ions in the feed solution would be accelerated to the production of a large amount of sulfuric acid and mixed base solutions, forming the low resistance of the stack. Thus, appropriately increasing the current should be beneficial to form a low voltage drop for the BMED system and shorten the operation time. Besides, it can be noticed clearly that whichever current density, the voltage of the stack has changed before and after adding conjugate salt. The addition of the (NH_4_)_2_SO_4_ solution, increasing the conductivity of feed solution, reduced the stack resistance. For an instance, the onset voltage decreased from 37.4 to 27.3 V after adding (NH_4_)_2_SO_4_ solution at the current density of 50 mA/cm^2^. Therefore, a low initial concentration of feed solution in the salt compartment was responsible for a relatively high resistance [18].

Figure 4. showed the effect of current density on the concentration of H_2_SO_4_ produced by the BMED stack. It can be seen in Figure 4a that the concentration of H_2_SO_4_ increased approximately linearly with time at a fixed current density. However, the increment between two adjacent H_2_SO_4_ concentrations is decreasing when the experiment was near to finished; the reason was that the back diffusion of ions through an ion exchange membrane caused by a high concentration gradient between the acid and salt compartments may hinder the transport of SO_4_^2−^ ions [19]. In Figure 4b, the H_2_SO_4_ concentration increased with the increase of current density, resulting from the higher migration rate of SO_4_^2−^ ions under the higher current density in the stack. In addition, the water dissociation rate was also accelerated with the increase of current according to the second Wien effect [20]. It also can be found in Figure 4b that there were no significant differences in acid concentration before and after the filling of (NH_4_)_2_SO_4_ when the current density was in the range of 30 and 40 mA/cm^2^, which was ascribed to the dissociation degree of water molecules and the migration rate of hydrate ions in the feed compartment being restricted at a low current density. When the current density exceeded 40 mA/cm, due to the synergistic effect of the applied current and the homogeneous current, the yield of H_2_SO_4_ was significantly increased after the addition of (NH_4_)_2_SO_4_. When the current density was 50 mA/cm^2^, the H_2_SO_4_ concentration increased from 0.87 to 0.95 mol/L.

To determine the optimum current density, the current efficiency and energy consumption were investigated with different current densitys for the acid compartment, and the results are shown in Figure 5. Whether with or without (NH_4_)_2_SO_4_, there was no obvious change in current efficiency with current densities between 30 and 50 mA/cm^2^, both the current efficiency and the acid concentration increased steadily. When the current density exceeded 50 mA/cm^2^, due to the penetration of salt ions through the bipolar membrane and protons leakage induced by the tunnel transport mechanism through the anion exchange membrane being accelerated [21,22], the current efficiency sighltly declined. When it came to the energy consumption, it increased almost linearly as the current density increased; the leakage of more proton ions could be responsible for this phenmomenon.

However, at a fixed current density, the current efficiency and energy consumption of the acid decreased slightly after introducting (NH_4_)_2_SO_4_, resulting from the extension of operating time and the reduction of resistance. Taking 50 mA/cm^2^ as an example, the current effiency decreased from 62.18% to 55.55%, and the energy consumption decreased from 3.04 to 2.42 kwh/kg after adding (NH_4_)_2_SO_4__._

### 3.2. Effect of Initial Ammonium Sulfate Concentration

According to the experiment above, the optimum current density was detemined, and the concentraton of the conjugate salt added in the Na_2_SO_4_ solution was also our priority. Therefore, the experiments implemented at 50 mA/cm^2^ were conducted with solution containing 15 wt % Na_2_SO_4_ and (NH_4_)_2_SO_4_, in which (NH_4_)_2_SO_4_ involved different mass fractions such as 2 wt %, 4 wt %, 6 wt %, 8 wt %, and 10 wt %, respectively. The effect of (NH_4_)_2_SO_4_ concentration added in the feed compartment on voltage, acid concentration, current efficiency, and energy consumption was investigated.

In Figure 6, the voltage of the BMED stack as time elapsed when adding different concentrations of (NH_4_)_2_SO_4_ was shown in Figure 6a. Obviously, the higher the concentration of (NH_4_)_2_SO_4_ in feed solution, the lower the electrical resistance of the membrane stack, resulting in the lower membrane stack voltage. For each curve, in the initial stage of each experiment, due to the acid and base compartment with deionized water, the stack electrical resistance was high. Along with the production of sulfuric acid and the mixed base, the resistance abruptly declined. Afterwards, the resistance remained invariable for a while, indicating that the process of ion migration was at a stable stage [23]. At the end of the experiments, the membrane stack resistance increased, which was ascribed to the depletion of electrolyte in feed solution. 

Figure 6b illustrated the change of H_2_SO_4_ yield when introducing different concentrations of (NH_4_)_2_SO_4_. When the concentration of (NH_4_)_2_SO_4_ was in the range of 2–8 wt %, the final acid concentration increased from 0.89 to 1.215 mol/L. However, the acid concentration declined instead (from 1.215 to 1.153 mol/L) when the mass fraction of (NH_4_)_2_SO_4_ was up to 10 wt %. According to the van’t Hoff equation, a chemical potential created by high osmotic pressure that was generated from a high concentration in feed solution was exactly opposite to the direct current field force and inhibited the ion migration from the salt compartment to acid and base compatments [24,25].

The current efficiency and energy consumption of the acid compartment with different concentrations of (NH_4_)_2_SO_4_ is shown in Figure 7. It can be seen that the trend of current efficiency change was similar to that of acid concentration; current efficiency was on the rise from 44.04% to 60.12% when the concentration of (NH_4_)_2_SO_4_ increased from 2 to 8 wt %. However, 10 wt % (NH_4_)_2_SO_4_ was added in feed solution including 15 wt % Na_2_SO_4_, and the current efficiency was a little less than that with 8 wt % (NH_4_)_2_SO_4_. The initial increase can be attributed to the combined probability of SO_4_^2−^ and H^+^ ions increasing. However, when the concentration of (NH_4_)_2_SO_4_ increased to a certain value, many more SO_4_^2−^ ions will transport across the AMV under a higher feed concentration due to an increase of the diffusion effect. However, the water-splitting rate was in a stable state under a fixed current, and a large number of H^+^ was produced. This also suggests that the molar ratio of H^+^ and SO_4_^2−^ was larger than 1. Thus, there were not enough SO_4_^2−^ ions combined with the water-splitting H^+^ ions. To maintain the charge conservation of the acid compartment, the redundant H^+^ ions would transport across the AMV, which would decrease the current efficiency [25,26]. The trend of energy consumption was just opposite, since the lower resistance of the stack will be with the higher concentration in feed solution, thereby decreasing the energy consumption accordingly [27].

As a result, a high current efficiency (60.12%) and low energy consumption (2.59 kWh/kg) were achieved for H_2_SO_4_ production by the BMED process with solutions containing 15 wt % Na_2_SO_4_ and 8 wt % (NH_4_)_2_SO_4_ at a current density of 50 mA/cm^2^. Furthermore, the mixture solutions of 15 wt % Na_2_SO_4_ 8 wt % (NH_4_)_2_SO_4_ would be used in further research studies.

### 3.3. Effect of Feed Solution Volume

The above experiments have determined the optimal current density and the dosage of (NH_4_)_2_SO_4_. Since the concentration of acid was also limited in the lab-scale apparatus when the feed solution in salt compartment was exhausted, it is thus necessary to further explore the effect of feed solution volume on the concentration of H_2_SO_4_ generated from the BMED process [28]. The experiments were conducted with the feed solution, in which the concentrations of Na_2_SO_4_ and (NH_4_)_2_SO_4_ were 15 wt % and 8 wt % respectively, the volume of both acid and base compartment were 500 mL, and the volumes of the feed solution were 500 mL, 1000 mL, and 1500 mL, respectively (the ratios of *V*_feed_ and *V*_acid_ were 1:1, 2:1, and 3:1, respectively); the other conditions were same as those in Section 3.2.

The change of concentration of H_2_SO_4_ produced, current efficiency, and energy consumption in the BMED process with the different volumes of feed solution are presented in Figure 8. It was shown from Figure 8a that the concentration of H_2_SO_4_ increased from 1.38 to 1.58 mol/L when the volume of salt solution increased from 500 to 1500 mL; this was because the increase of salt solution, according to the Moore’s law, represented the higher total mole number of feed solution, which was beneficial for more SO_4_^2−^, Na^+^, and NH_4_^+^ ions to migrate into the acid and base compartments [28].

The current efficiency and energy consumption of the acid produced in the acid compartment with the different volumes of feed solution is shown in Figure 8. It was found from Figure 8b that the increase of salt volume could improve the current efficiency and reduce the energy consumption of the acid. The current efficiency of acid increased from 68.03% to 77.93%; meanwhile, the energy consumption of generating acid decreased from 1.76 to 1.56 kwh/kg when the volume of salt increased from 500 to 1500 mL. The increase of current efficiency was ascribed to the acid increment increasing over the same time period.

Although the stack resistance rose slightly, the leakage of ions was alleviated, resulting in the decrease of energy consumption. Therefore, under the premise of adding a certain concentration of conjugate salt, increasing the volume of feed solution may be also a good way to improve the concentration of acid.

### 3.4. Effect of Membrane Stack Configuration

The production yield of acid by the BMED process was not only related to the operation parameters mentioned above, but it was also associated with the membrane stack configuration. In order to investigate the effect of membrane stack configuration on the production of H_2_SO_4_, a two-compartment stack was applied in a further experiment; its configuration is shown in Figure 1b, in which there were only anion exchange membranes and bipolar membranes that arranged alternately in the membrane stack when the cation exchange membranes were removed. In the two-compartment of a BMED system, pure acid and the mixed solution of salt and base were produced in the acid and salt compartments, respectively. Voltage change, the production of H_2_SO_4_, current efficiency, and energy consumption with different membrane stack configurations were discussed.

Figure 9 presented the change of the voltage as time filed with the different stack configurations. At the beginning, the initial voltage in the BP-A-C-BP (“BP” means bipolar membrane, “A” means anion exchange membrane, and “C” means cation exchange membrane, “BP-A-C-BP” means that two bipolar membranes, an anion exchange membrane, and a cation exchange membrane are alternately arranged to form a repeating unit of the membrane stack) configuration was slightly higher than that in the BP-A-BP configuration. This was because the solutions in the adjacent chambers of the bipolar membrane were deionized water and feed solution, respectively, in a BP-A-BP configuration. When the current was passed, the BMED system was in a non-ideal state, and the migration of salt ions was prior to water dissociation, resulting in the higher voltage. As the experiment continued, the experiment reached a steady state, and the voltage of the BP-A-C-BP configuration tended to be higher than that of the BP-A-BP configuration. At the end of the experiment, the voltage developed in a stable direction in the BP-A-BP configuration, while in the BP-A-C-BP configuration, the voltage reflected a rising trend because of the high concentration of H_2_SO_4_ in the acid compartment [29].

Figure 10 listed the change of sulfuric acid concentration, current efficiency, and energy consumption in two kinds of stack configurations. It was found from Figure 10a that the concentration of H_2_SO_4_ was higher in the BP-A-C-BP configuration than that in the BP-A-BP configuration. In the BP-A-BP configuration, the concentration of H_2_SO_4_ was equal to twice the number of moles of hydrogen produced in the acid compartment subtracted by the acid loss caused by the competition with the hydroxide ions; these hydroxide ions generated from the salt compartment could pass across the anion exchange membrane and neutralize with hydrogen ions. However, the feed concentration played a buffer role in the BP-A-C-BP configuration, which reduced the undesirable influence [29,30].

As shown in Figure 10b, the current efficiency decreased with time, which conformed to the typical trends, thereby restricting the possibility of obtaining a high concentration of H_2_SO_4_. In addition, compared with the current efficiency in the BP-A-C-BP configuration, that in the BP-A-BP configuration was lower. In a three-compartment system, neither of the other cations compete with H^+^ nor anions with sulfuric ions, so the current efficiency was relatively high. For a two-compartment system, the current efficiency of acid decreased by the water transport from the salt compartment to the acid compartment because of the thinner of cation exchange membrane. On the other hand, due to the size of the OH^−^ being smaller than the sulfuric ions, the OH^−^ diffusion speed through the membrane would be larger than that of the sulfate ions, which was also responsible for the decreased current efficiency [31].

On the contrary, the energy consumption in the BP-A-BP configuration was slightly higher than that in the BP-A-BP configuration. The energy consumption of acid was determined by not only the stack resistance but also the concentration of H_2_SO_4_ produced in the acid compartment, as shown in Equation (2). Thus, the higher energy consumption obtained in the BP-A-BP configuration resulted from the low conductivity of the H_2_SO_4_ solution [32].

## 4. Conclusions

The feasibility study of improving the concentration of sulfuric acid by adding conjugate salt in a lab-scale BMED system with simulated sodium sulfuric wastewater was proven in this work. The results presented that the current density, concentration of ammonium sulfate, feed solution volume, and membrane stack configuration could affect the yield of sulfuric acid, current efficiency, and energy consumption in the BMED system. It was shown that the higher current density and ammonium sulfate concentration were in favor of producing a higher concentration of H_2_SO_4_. When adding 8 wt % (NH_4_)_2_SO_4_ to feed solution including 15 wt % Na_2_SO_4_ at 50 mA/cm^2^, the concentration of H_2_SO_4_ increased from 0.89 to 1.215 mol/L. Simultaneously, the higher current efficiency (60.12%) and lower energy consumption (2.59 kWh/kg) were also obtained. Besides, when the concentration of the homologous added in the feed compartment was determined, the volume of feed solution was also a vital operating parameter that affected the acid concentration. When the volume of feed solution increased from 500 to 1500 mL, the concentration of H_2_SO_4_ increased from 1.38 to 1.58 mol/L. Furthermore, compared with the two-compartment BMED system, it is more appropriate for a three-compartment BMED system to produce a higher concentration of acid.

Therefore, it is an environmentally friendly and energy-saving method to increase sulfuric acid concentration by adding conjugate salt in the treatment of sodium sulfate wastewater in a BMED system. The method was also suitable for other salt-containing wastewaters and can be reproducibly produced in industrial processes.

## Figures and Tables

**Figure 1 polymers-12-00343-f001:**
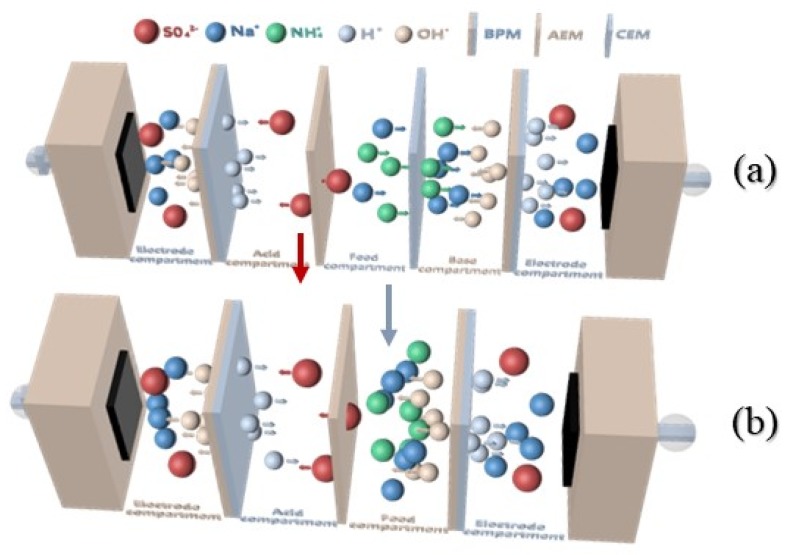
Schematic representation of bipolar membrane electrodialysis (BMED) configuration with three (**a**) and two compartments (**b**).

**Figure 2 polymers-12-00343-f002:**
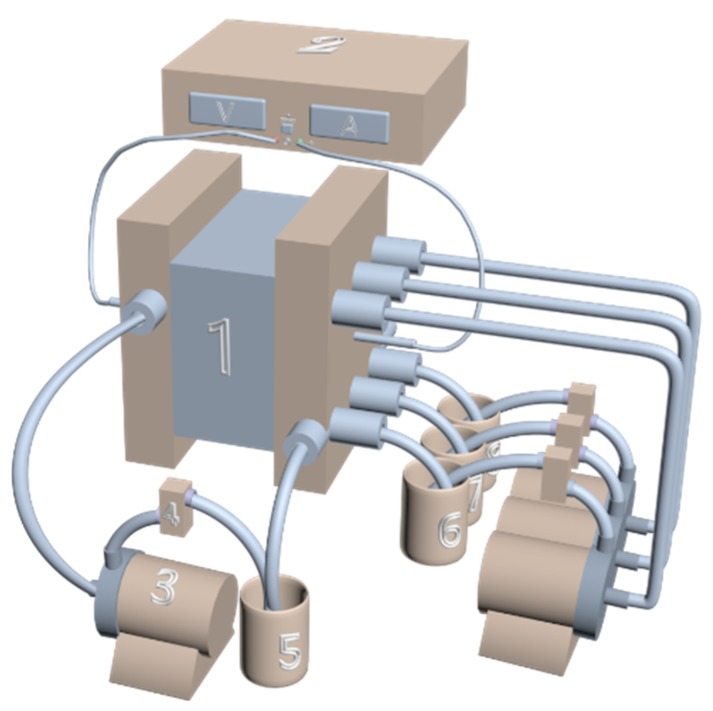
Schematic diagram of BMED system. (**1**) BMED system stack; (**2**) DC power supply; (**3**) centrifugal pump; (**4**) rotameter; (**5**) electrode rinse tank; (**6**) base solution tank; (**7**) feed solution tank; (**8**) acid solution tank.

**Figure 3 polymers-12-00343-f003:**
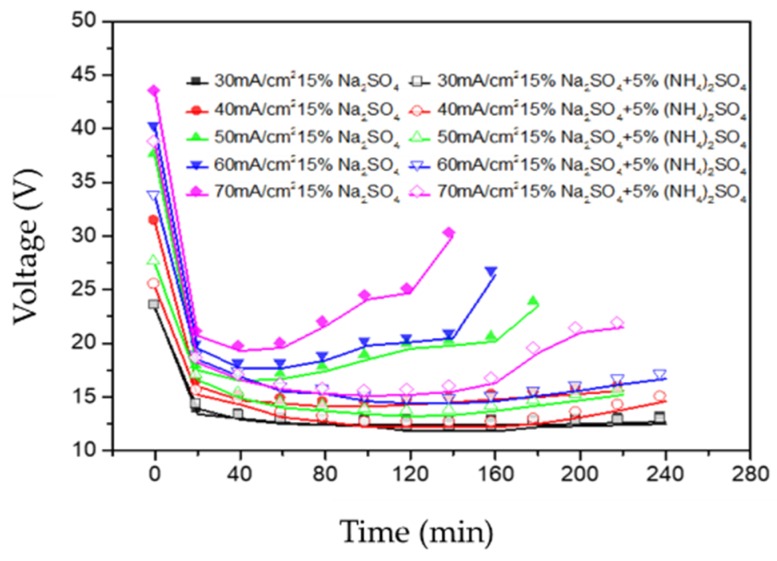
The voltage–time curves before and after adding (NH_4_)_2_SO_4_ with different current densities.

**Figure 4 polymers-12-00343-f004:**
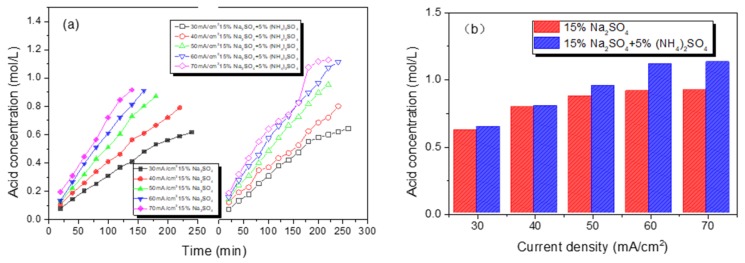
Production of H_2_SO_4_ (**a**) before and after adding (NH_4_)_2_SO_4_ with different current densities (**b**).

**Figure 5 polymers-12-00343-f005:**
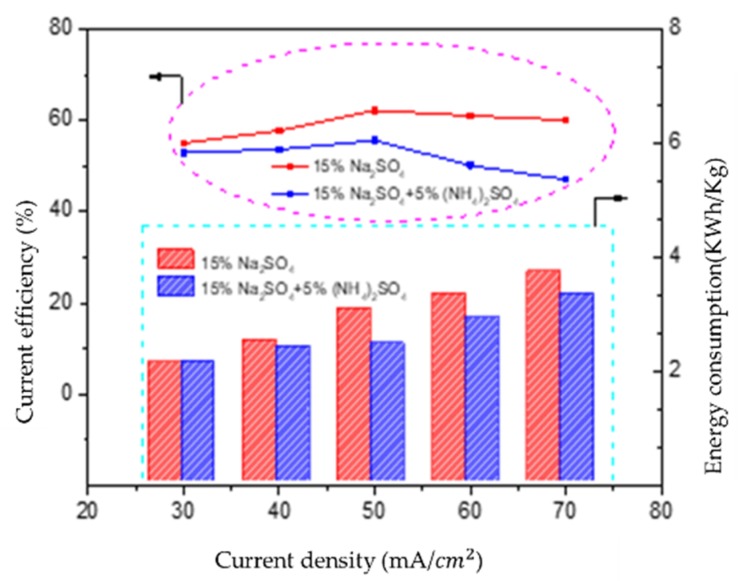
The change of current efficiency and energy consumption before and after adding (NH_4_)_2_SO_4_ with different current densities.

**Figure 6 polymers-12-00343-f006:**
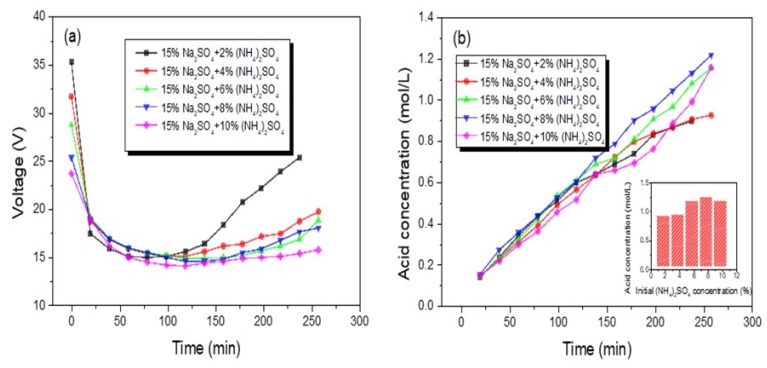
The voltage–time curves (**a**), the acid concentration–time curves (**b**) in the BMED process with different mass fractions of (NH_4_)_2_SO_4_ and the acid concentration-initial (NH_4_)_2_SO_4_ concentration relationship.

**Figure 7 polymers-12-00343-f007:**
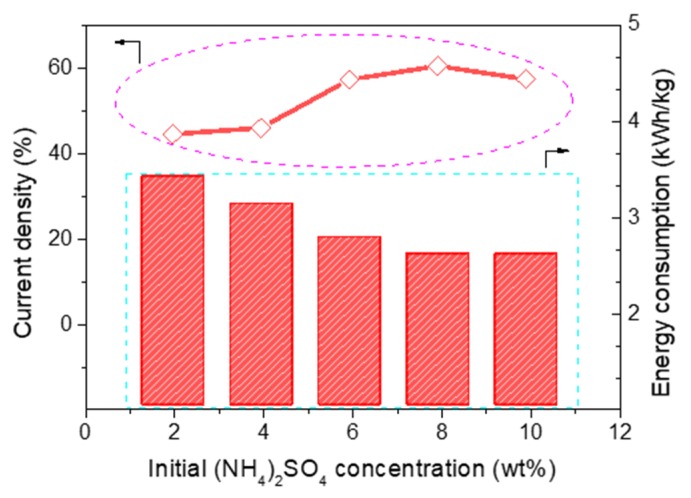
The change of current efficiency and energy consumption with different concentrations of (NH_4_)_2_SO_4__._

**Figure 8 polymers-12-00343-f008:**
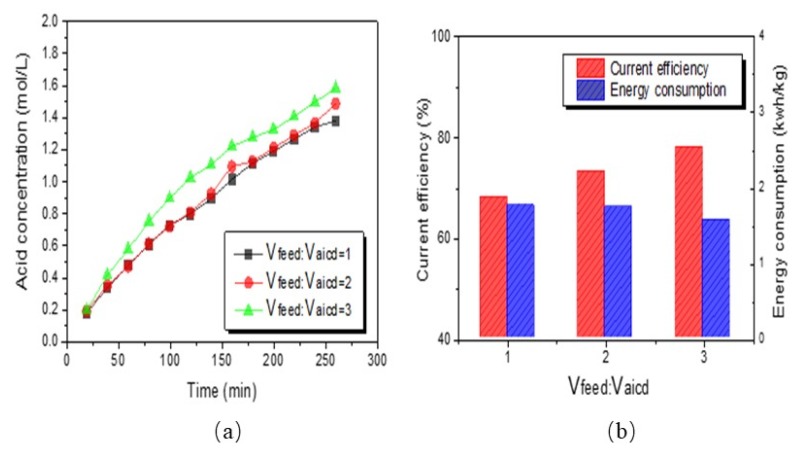
The effect of feed solution volume on the acid concentration (**a**) and current efficiency and energy consumption (**b**).

**Figure 9 polymers-12-00343-f009:**
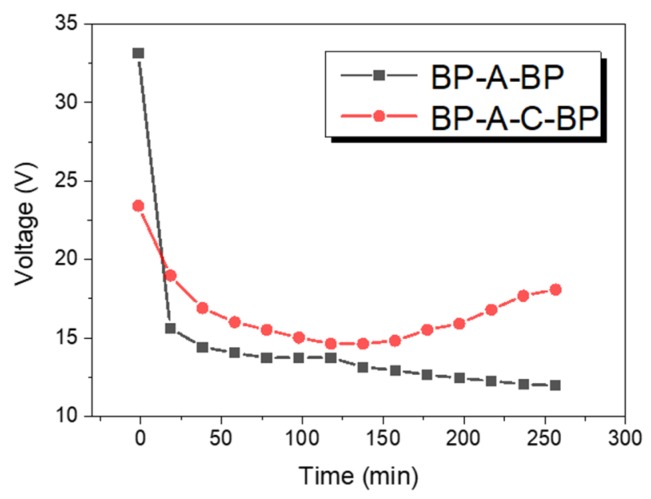
The voltage drop across the stack in different configurations.

**Figure 10 polymers-12-00343-f010:**
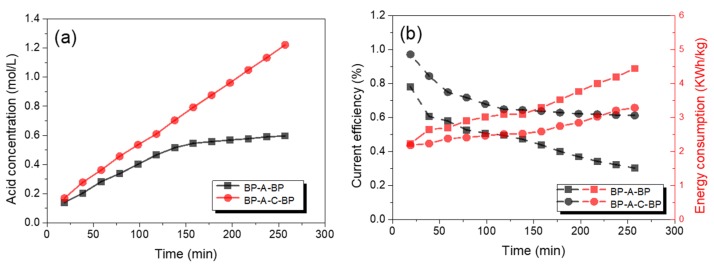
The concentration of sulfuric acid (**a**), current efficiency, and energy consumption (**b**) in different configurations.

**Table 1 polymers-12-00343-t001:** The main properties of the membrane during the experiment.

	CMV	AMV	NEOSEPTA BP-1E
Exchange capacity (eq/kg)	2.4	1.9	-
Thickness (mm)	0.15	0.14	0.22
Resistance (Ω·cm^2^)	0.3–3	2.8–7	-
Burst strength (MPa)	0.2	0.2	≥0.40
Transport number	0.98	0.96	-
Water splitting voltage ^a^	-	-	1.2V ^b^
Water splitting efficiency ^a^	-	-	≥0.98

^a^ 1 N NaOH and 1 N HCl 10 A/dm^2^ 30 °C. ^b^ Potential difference measured between silver-silver chloride electrodes.

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
