# Peer review of "The Novel Strategy for Increasing the Efficiency and Yield of the Bipolar Membrane Electrodialysis by the Double Conjugate Salts Stress"

_polymers, 2020, doi:10.3390/polym12020343_

Round 1
Reviewer 1 Report
The comments are in the attached file.

Reviewer 2 Report
The authors have performed an investigation on the bipolar membrane electrodialysis for waste water treatment and acid recovery. The work scientifically sounds good but a major revision is required before it gets published:
Get help of language check by a native speaker. Take care of the formats, e.g. Page 1, line 20: subscripts in “H2SO4”. Page 1, line 29: Define “BP-A-C-BP”. Select a concise key words. The authors might improve the introduction. Please have a look at the following papers: i) C.A. Quist-Jensen, F. Macedonio, D. Horbez, E.J.D. Drioli, Reclamation of sodium sulfate from industrial wastewater by using membrane distillation and membrane crystallization, 401 (2017) 112-119. ii) R.A. Tufa, J. Hnát, M. Němeček, R. Kodým, E. Curcio, K. Bouzek, Hydrogen production from industrial wastewaters: An integrated reverse electrodialysis - Water electrolysis energy system, Journal of Cleaner Production, 203 (2018) 418-426. Page 2, line 43: “Bipolar membrane electrodialysis (BMED) can dissociate hydrolysis….” Is not a right saying, so adjust. Page 2, line 46: Take care of the full stop. Improve the font of texts, legends in Fig. 1, 3, 4a, 5 Experimental part: Which type of spacer material is used in the BMED? Define the BP-A-BP configuration. Page 6, line 185: Increase in the water dissociation rate at high currents due to the second Wien effect improves the yield of H2SO4. What is the potential negative impact of high current density in your system? Check typos, e.g., Page 6, line 201: “throngh” should be “through” How does the two configurations BP-A-C-BP and BP-A-BP compare in terms of Ohmic losses. If possible, plote all the factors influencing energy consumption and justify your argument that BP-A-C-BP configuration consumes less energy than BP-A-BP.

Round 2
Reviewer 2 Report
The manuscript is now improved. A small suggestion is to check the English, typos again as some references are not displayed in the manuscript. Error showing.